Analysis of the chitin synthase gene family in Ganoderma lucidum: its structure, phylogeny, and expression patterns

Liu Linling
Yang Yiming
Li Jintao
Gao Yanliang
Yan Meixia yanmeixia@caas.cn
Institute of Special Economic Animal and Plant Sciences, Chinese Academy of Agricultural Sciences , Changchun , Jilin , China
Nunes-da-Fonseca Rodrigo
Electronic publication date: 2025 Nov 5
Publication date: 2025
Volume: 13
Electronic Location ID: e20302
Received 2025 Jul 9; Accepted 2025 Oct 7
Copyright: ©2025 Liu et al.
Copyright year: 2025
Copyright holder: Liu et al.
License: This is an open access article distributed under the terms of the Creative Commons Attribution License, which permits unrestricted use, distribution, reproduction and adaptation in any medium and for any purpose provided that it is properly attributed. For attribution, the original author(s), title, publication source (PeerJ) and either DOI or URL of the article must be cited.
License URL: https://creativecommons.org/licenses/by/4.0/

Keywords: Chitin synthase gene family, Ganoderma lucidum, Genome-wide identification, Bioinformatics, Expression analysis

Funding: Jilin Province Science and Technology Development Program Project YDZJ202401455ZYTS This work was supported by the Jilin Province Science and Technology Development Program Project (YDZJ202401455ZYTS). The funders had no role in study design, data collection and analysis, decision to publish, or preparation of the manuscript.

==============================
Background

Chitin synthases are essential enzymes in fungi, contributing to various biological processes such as hyphal growth, sporulation, and cell wall stability. Despite their well-documented functions in other fungal species, the specific roles of chitin synthases in Ganoderma lucidum remain unexplored. This investigation systematically characterized the complete chitin synthase gene family in Ganoderma lucidum.

Methods

A comprehensive analysis was conducted using bioinformatics tools to examine genomic localization, gene structure, conserved domains, and phylogenetic evolution. By employing bioinformatic approaches, the study investigated promoter cis-elements and expression patterns to elucidate the regulatory mechanisms of these genes in G. lucidum.

Results

In G. lucidum, eight chitin synthase (GlCS) family members were identified and phylogenetically classified into five distinct classes. Our investigation revealed stage-specific expression patterns of GlCS genes throughout the fungal development process. GlCS1, GlCS2, GlCS5, GlCS6, and GlCS8 exhibited significantly greater expression levels in the early fruiting body (EA) stage than in the other developmental phases. In the mature fruiting body (MA) stage, GlCS3 was predominantly expressed. In the primordium formation (PR) stage, GlCS7 exhibited peak expression levels. Six genes (GlCS1, GlCS3, GlCS4, GlCS6, GlCS7, and GlCS8) were markedly upregulated under 40 °C thermal stress, suggesting their potential roles in thermotolerance mechanisms. These findings demonstrate functional diversification among GlCS family members across different developmental stages and stress conditions.

Introduction

The fungal cell wall (FCW) is a highly dynamic and multifunctional structural system that provides essential mechanical support and morphological plasticity to fungal cells. More importantly, the chemical composition of the cell wall is closely linked to the regulation of fungal growth and development. Although compositional variations exist among different fungal species, the dry weight composition of the cell wall is remarkably conserved: polysaccharides account for approximately 90% (with β-glucans comprising 50–60% and chitin comprising 10–20%), whereas glycoproteins make up 20–30% of the total (Latgé, 2007; Bernard & Latgé, 2001). As a core structural component, chitin creates intricate cross-linked networks with glucans, forming a multilayered skeletal cell wall framework (Bowman & Free, 2006). This architecture is indispensable for maintaining the integrity and mechanical properties of the cell wall (Bulawa, 1993). Fungi are capable of converting various carbon sources—including glucose, trehalose, and glycogen—into chitin. Although this biosynthetic process involves a multistep enzymatic reaction network, its core pathway remains highly conserved across different fungal species (Chen et al., 2022). Among these enzymes, chitin synthase acts as the key rate-limiting enzyme not only by precisely regulating the spatial localization process of chitin synthesis but also by directly determining its efficiency. This ensures that chitin production is predominantly concentrated in actively growing fungus regions, such as budding sites and hyphal tips (Merzendorfer, 2011; Bracker, Ruiz-Herrera & Bartnicki-Garcia, 1976).

Since the first chitin synthase (CS) gene was successfully cloned from Saccharomyces cerevisiae in 1986, remarkable progress has been made in terms of understanding the structural and functional diversity of this enzyme family (Bulawa et al., 1986). CS proteins are characterized by multiple transmembrane helical domains and exhibit significant structural and functional variations. A phylogenetic analysis conducted based on amino acid sequences categorized chitin synthases into seven evolutionary classes (I–VII), although controversies persist among different classification systems (Mandel et al., 2006; Ruiz-Herrera & Ortiz-Castellanos, 2010). A widely accepted model proposed by Choquer et al. (2004) suggests a two-branch evolutionary framework: the first branch is composed of classes I, II, and III, whereas the second branch includes classes IV, V, and VI, with class VII forming an independent ancestral lineage. A structural domain architecture analysis reveals critical distinctions among these classes. Classes I–III include conserved CS1 catalytic domains (PF01644) and N-terminal regulatory regions (CS1N, PF08407), which form canonical transmembrane catalytic complexes (Riquelme & Bartnicki-García, 2008). In classes IV–VII, the core structural domain is CS2 (PF03142) (Gonçalves et al., 2016). Classes IV–VI additionally contain cytochrome b5-like heme-binding domains (cyt-b5, PF00173), whereas classes V and VI possess N-terminal myosin motor domains (MMD, PF00063), suggesting potential links to cellular motility or vesicle trafficking (Li et al., 2016; Liu et al., 2017). Class VII retains only the CS2 basal domain, supporting its hypothesized role as the evolutionary prototype of the family. In the filamentous Aspergillus nidulans fungal model, eight CS family members have been identified (AnChsA–G, AnCcsmA, and AnCsmB), and they display distinct phylogenetic distributions: classes II (AnChsA), III (AnChsB), I (AnChsC), IV (AnChsD), III (AnChsE), VI (AnChsF), V (AnCsmB), and VII (AnCcsmA). This noncontiguous classification pattern reflects the coevolution of CHS functional diversification with fungal morphological complexity (Motoyama et al., 1996). Chitin synthases play indispensable roles in maintaining critical physiological and biochemical processes in fungi, including normal growth and development, sporulation, and cell wall integrity (Horiuchi, 2009). In S. cerevisiae, three chitin synthases (CS1, CS2, and CS3) perform distinct yet coordinated cell wall maintenance and cytokinesis functions (Roncero, 2002). CS1 primarily exists as an inactive zymogen that requires proteolytic activation (Bulawa et al., 1986; Cabib, 1989). During the cell division process, it participates in primary septum formation and cooperates with CS2 during cell separation. CS2 serves as the principal enzyme that is responsible for synthesizing the primary septum—a crucial structure for daughter cell detachment (Silverman et al., 1988; Shaw et al., 1991). Although its contribution to total chitin production is minimal, it is essential for properly performing cytokinesis. CS3, which is nonessential for viability, accounts for approximately 90% of cellular chitin production (Roncero et al., 1988; Valdivieso et al., 1991). Unlike CS1 and CS2, it localizes chitin to the bud neck and cell wall to reinforce structural stability rather than directly participating in septum formation. Whereas CS1 and CS2 are primarily involved in cell division, CS3 ensures a widespread chitin distribution. This functional specialization enables the concerted maintenance of cell wall robustness and division fidelity.

Environmental stress factors, including temperature extremes, humidity fluctuations, light exposure, and CO2 levels, critically influence the growth and secondary metabolism of G. lucidum. Among these, high-temperature stress is a key limiting factor, as it disrupts mycelial viability, suppresses polysaccharide biosynthesis, and triggers oxidative damage. Microorganisms exhibit diverse responses to heat stress that significantly affect their growth and metabolism. While heat exposure triggers cell death in S. cerevisiae (Guyot et al., 2015), Metarhizium robertsii develops tolerance through pyruvate accumulation despite suppressed mycelial growth (Zhang, St. Leger & Fang, 2017). Similarly, G. lucidum demonstrates temperature sensitivity, with growth completely ceasing at 40 °C (Zhang et al., 2021).

G. lucidum, which is a traditional medicinal and edible fungus, requires the coordinated regulation of multiple genes for proper fruiting body development (Seweryn, Ziała & Gamian, 2021). As an essential regulatory element in fungal biology, chitin synthase (CS) governs key processes such as morphogenesis of fruiting bodies and cellular stress reactions. Based on the G. lucidum genome, we comprehensively identified and analyzed the CS genefamily through bioinformatics approaches. We investigated transcriptomic data to explore the expression patterns of this gene family during fruiting body development and under heat stress, which provided deeper insights into the functional roles of chitin synthase genes. The reliability of the gene expression data was further confirmed by quantitative real-time PCR (qPCR), supporting future investigations and potential applications. Determining the functions of chitin synthase genes in G. lucidum with molecular biology techniques will provide a scientific basis for the cultivation of high-quality strains.

Materials and Methods

Identification of Chitin synthase genes

The genomic data of G. lucidum were obtained from the JGI Mycocosm database (available at  https://mycocosm.jgi.doe.gov/Ganluc1/Ganluc1.home.html) (Chen et al., 2012). Among fungal chitin synthases, the CHS1, CHS2, and CHS3 genes from Saccharomyces cerevisiae represent the most well-characterized members. These three genes have distinct functions: CHS1 participates in cell wall repair during cytokinesis, CHS2 is responsible for primary septum formation between mother and daughter cells, and CHS3 forms the bud scar ring where the majority of cell wall chitin is deposited. Multiple classes of chitin synthase genes have been identified in other fungal species. Based on sequence similarity, these genes are primarily classified into two major groups: the majority showing homology to CHS1 and CHS2 are categorized as classes I–III chitin synthases, while the remainder, exhibiting high similarity to CHS3, are classified as classes IV and V (Henar Valdivieso, Durán & Roncero, 1999). Three chitin synthase (CS) sequences from S. cerevisiae (NP_009594.1, NP_014207.2, and NP_009579.1) were retrieved from the NCBI database and subjected to a local BLAST analysis against the G. lucidum genome database for sequence alignment purposes (Altschul et al., 1990). The E value was less than 1e−5 in the BLAST search. The putative G. lucidum CS genes were further validated by performing an online BLAST search against the NCBI NR database. The conserved Chitin_synth domain in the G. lucidum CS gene protein was identified with domain analysis tools (Wang et al., 2023).

Bioinformatics analysis

The ExPASy ProtParam tool (Hallgren et al., 2022) was used to assess various physicochemical parameters of the G. lucidum CS protein. For subcellular localization prediction, the WoLF PSORT algorithm was utilized. Membrane-spanning regions were identified through TMHMM Server v. 2.0, while secondary protein structure elements such as α-helices (Geourjon & Deleage, 1995), β-strands, and unstructured loops were determined via the SOPMA prediction platform (Letunic & Bork, 2024).

To investigate the evolutionary relationships between GlCS genes from G. lucidum and CS genes from other fungi, we retrieved CS gene sequences of the following species from the NCBI database: Saccharomyces cerevisiae (NP_014207.2, NP_009594.1, NP_009579.1), Pleurotus ostreatus (BAF76742.1, KAF7436169.1, XP_036626338.1), Lentinula edodes (BAJ08814.1, QYK92509.1), Agaricus bisporus (CAA67114.1), Leucoagaricus  sp. SymC.cos (KXN87631.1, KXN85737.1, KXN85065.1, KXN90875.1, KXN90206.1, KXN88901.1), and Aspergillus nidulans (XP_050467923.1, XP_050467691.1, XP_661971.1, BAA11845.1, AAB05797.1). A phylogenetic tree was then constructed based on these sequences for further analysis. The amino acid sequences were aligned in MEGA11 (Gleeson et al., 2014), and this was followed by phylogenetic reconstruction via the neighbor-joining (NJ) method. A bootstrap analysis with 1,000 replicates was conducted, while all other parameters remained at their default values. The resulting phylogenetic tree was further optimized and graphically presented with the interactive Tree of Life (iTOL) online platform to improve the visualization and interpretation processes (https://itol.embl.de/) (Duvaud et al., 2021). A motif analysis of the chitin synthase (CS) genes in G. lucidum was performed with MEME (https://bio.tools/meme_suite) (Tamura, Stecher & Kumar, 2021). Default search parameters were employed for motif identification. On the basis of the genomic annotations of G. lucidum, a comprehensive gene structure analysis was conducted on the CS family members with TBtools v2.119 software (Chen et al., 2020). This analysis included an examination of exon–intron structures, conserved motifs, and functional domains to elucidate the evolutionary relationships and functional diversity within the CS gene family.

Using the Fasta Extract tool in the TBtools v2.119 software (Chen et al., 2020), the promoter sequences 2,000 bp upstream of the start codon of the CS family genes in G. lucidum were extracted. Cis-acting element analysis was then performed using the PlantCare database (Lescot et al., 2002).

Expression patterns of the GlCS genes during different developmental stages

The experimental strain used in this study was G. lucidum dikaryotic strain LZ8, a species cultivated in Jilin Province, China. This strain is maintained by the Edible and Medicinal Mushroom Research Group at the Institute of Special Animal and Plant Sciences, Chinese Academy of Agricultural Sciences. For mycelial (MY) stage samples, strain LZ8 was cultured on potato dextrose agar (PDA) medium for 10 days, and samples were collected after the plates were fully colonized. Fruiting body samples, including the primordium (PR), early fruiting body (EA), and mature fruiting body (MA) stages, were provided by the Yimutian Agriculture Company. A schematic of the sample is presented in Fig. 1. As shown in Fig. 1, all samples were collected from healthy growing mycelial and fruiting body tissues where cells remained intact with fully preserved cell walls.

Figure 1 Different developmental stages of Ganoderma lucidum.

Immediately after collection, the samples were chilled in liquid nitrogen and subsequently preserved in a −80 °C freezer. All of the samples were tested in triplicate, and the experiments were performed on three biological replicates. The mycelial (MY) stage samples were used as the control group for comparative purposes.

Expression patterns of the GlCS genes under heat stress

For mycelial (MY) stage samples, strain LZ8 was cultured on potato dextrose agar (PDA) medium for 10 days at 25 °C. For the heat stress samples, after the mycelial (MY) stage samples were cultured on potato dextrose agar (PDA) medium for 10 days at 25 °C, they were transferred to 40 °C and incubated for 6 h, serving as the mycelial at 40 °C (MY-40) samples.

Immediately after collection, the samples were chilled in liquid nitrogen and subsequently preserved in a −80 °C freezer. All of the samples were tested in triplicate, and the experiments were performed on three biological replicates. The mycelial (MY) stage samples at 25 °C were used as the control group for comparative purposes.

RNA extraction and RT–qPCR analysis

Total RNA isolation was performed using a commercial RNA extraction kit (Nanjing Vazyme Biotech) following the standard protocol. The kit’s proprietary solutions, including buffer RWA and RWB, effectively eliminated genomic DNA, phenolic compounds, and polysaccharide contaminants. RNA quality control was conducted using a Biochrom Bio Drop Duo spectrophotometer for concentration and purity measurement, supplemented by 1% agarose gel electrophoresis for integrity verification. The obtained RNA samples exhibited A260/280 ratios between 1.89 and 1.99, confirming satisfactory purity. Subsequently, 800 ng of purified RNA from each sample was subjected to reverse transcription using a dedicated cDNA synthesis kit (Nanjing Vazyme Biotech) to generate first-strand cDNA.

Gene-specific primers targeting the G. lucidum GlCS genes were designed on the basis of complete CDS sequences using Primer Premier 5.0, with 18S rRNA serving as a control (Table 1). All primers were commercially synthesized by Sangon Biotech (Shanghai). All primer efficiencies were tested.

Quantitative PCR was carried out in 20-µL reactions containing 10 µL of 2 ×Taq PCR Master Mix, 0.5 µL each of 10-µmol/L primers, one µL cDNA template, and eight µL of ddH2O with SYBR Green Super Mix (Beijing TransGen Biotech). The thermal cycling conditions were as follows: 95 °C for 5 min, and 40 cycles at 95 °C for 10 s, 60 °C for 10 s, and 60 °C for 10 s (fluorescence acquisition).

The expression levels were normalized by the 2−ΔΔCt method (Livak & Schmittgen, 2001). A data analysis was performed in Microsoft Excel, with the statistical evaluation and graphical presentation steps conducted in GraphPad Prism 9.5. We analyzed RT–qPCR data for 8 GlCS genes during different developmental Stages using an ANOVA. We analyzed the RT–qPCR data of 8 GlCS genes under heat stress using t tests.

Results

Identification of CS genes in Ganoderma lucidum and protein characterization

We performed a local BLAST search of the G. lucidum genome using the three chitin synthase (CS) proteins from S. cerevisiae as queries, which identified eight homologous CS genes. The results were further validated through an online BLAST comparison with the NCBI NR database, which led to the identification of eight CS sequences, designated as GlCS1 to GlCS8.

A domain analysis was conducted on the amino acid sequences of the eight CS genes with online bioinformatics tools. The results (Table 2) demonstrated that GlCS2, GlCS4, and GlCS5 contain three conserved domains: Chitin_synth_1, Chitin_synth_1N, and Chitin_synth_2. GlCS1, GlCS3, GlCS6, and GlCS8 were found to possess the Chitin_synth_2 domain, whereas GlCS7 contains only the Chitin_synth_1N domain. These findings highlight the structural diversity and functional complexity of the CS gene family in G. lucidum.

Table 1 Primer sequences.

Gene name	Forward primer sequence (5′–3′)	Reverse primer sequence (5′–3′)	Product size	
18S-RNA	TATCGAGTTCTGACTGGGTTGT	ATCCGTTGCTGAAAGTTGTAT	162	
GlCS1	GATAGCGAACAAGCGGCAGTCAT	GGGAGACAGGATACACAACAAAACA	114	
GlCS2	AAGGACGGGAAGGAGGTGATAGA	CAAATGCTGCTTGACCCAGTTCG	255	
GlCS3	ATCGCAGGTTGTGCTTTATGTATCC	CATCCGTCCAAATGTCAATCTACC	173	
GlCS4	ACGGAGCAGGAAGACTAATACAAAGG	GGCGACAGAGAAAAGGATGAAAG	192	
GlCS5	GTGAACATCATCTACGAAGAGCCG	TGAAGGTATCGTTGACAGAGAAAGTG	226	
GlCS6	ACGACACTATCCTATCCGTCCTAAAA	AACCGCTTGTGACTCTACTCTTATCC	263	
GlCS7	CCCACTCCAAACCAAAGCATAAC	ATAGCCTGTTCCACACGCATTCA	111	
GlCS8	GACTCGCAACTGACAAACCAAAT	CAATACACTCCTGCTCTCCTCAAG	213	

Table 2 The conserved domains.

Gene name	Feature domain	
GlCS1	Chitin_synth_2	
GlCS2	Chitin_synth_1, Chitin_synth_1N, Chitin_synth_2	
GlCS3	Chitin_synth_2	
GlCS4	Chitin_synth_1, Chitin_synth_1N, Chitin_synth_2	
GlCS5	Chitin_synth_1, Chitin_synth_1N, Chitin_synth_2	
GlCS6	Chitin_synth_2	
GlCS7	Chitin_synth_1N	
GlCS8	Chitin_synth_2	

The GlCS proteins displayed considerable size variations (Table 3), with polypeptide lengths spanning 860–1,967 amino acid residues and predicted molecular masses ranging from 97 to 219 kilodaltons (kDa). The isoelectric points (pIs) of these proteins ranged from 6.34 to 8.87, indicating diverse physicochemical properties. Transmembrane helix predictions revealed that these proteins typically contain five to seven transmembrane helices. Subcellular localization predictions revealed that all identified GlCS proteins exhibited a 100% probability of plasma membrane localization. Notably, four isoforms (GlCS1, GlCS4, GlCS6, and GlCS8) presented additional endoplasmic reticulum-targeting signals. Furthermore, GlCS3 and GlCS6 demonstrated potential nuclear localization signals, whereas GlCS6 exhibited unique dual-targeting potential to both peroxisomes and the mitochondrial inner membrane. Interestingly, GlCS8 was specifically predicted to localize to the vacuolar compartment.

Table 3 Basic information and subcellular localization properties of the CS genes.

Gene name	Genomic position	Number of amino acids	Relative molecular weight (kDa)	pI	TMHs	PSORT predictions	
GlCS1	scaffold_7: 719513-723436	1,140	124.737	8.87	5	plas: 24, E.R.: 2	
GlCS2	scaffold_1: 3740112-3743941	860	97.423	8.66	7	plas: 27	
GlCS3	scaffold_1: 4333259-4338223	1,457	161.81	8.77	6	plas: 24, nucl: 1	
GlCS4	scaffold_25: 594362-598922	929	104.28	6.48	7	plas: 24, E.R.: 1	
GlCS5	scaffold_27: 547881-552042	1,082	121.36	8.55	7	plas: 25	
GlCS6	scaffold_6: 1610310-1616687	1,967	218.32	5.68	6	plas: 20, E.R.: 2, nucl: 1, mito: 1, pero: 1	
GlCS7	scaffold_9: 1489668-1489668	885	99.81	8.23	7	plas: 27	
GlCS8	scaffold_12: 995721-1002922	1,911	214.95	6.34	6	plas: 18, E.R.: 4, vacu: 3	

A secondary structural analysis of the GlCS proteins revealed that random coils and alpha helices dominate their structures, with smaller proportions of beta turns and extended strands (Table 4). Specifically, the proportions of alpha helices among the GlCS gene family members ranged from 21.96% to 39.3%, indicating significant variability. The extended strands accounted for 12.87% to 15.12% of the structures, with minimal variations. Beta turns constituted 3.66% to 5.23% of the structures, also exhibiting few differences. In contrast, the percentages of random coils ranged from 40.35% to 60.47%, indicating considerable variability among the proteins.

Table 4 Analysis of secondary structure of the CS gene proteins.

Gene name	Alpha helix %	Extended strand %	Beta turn %	Random coil %	
GlCS1	28.33	14.56	4.57	52.54	
GlCS2	39.3	15.12	5.23	40.35	
GlCS3	21.96	13.8	3.77	60.47	
GlCS4	36.49	14.64	4.95	43.92	
GlCS5	32.07	15.06	4.53	48.34	
GlCS6	37.52	13.01	3.91	45.55	
GlCS7	38.53	13.45	4.86	43.16	
GlCS8	38.88	12.87	3.66	44.58	

Gene structures, protein motifs, and phylogenetic relationships

As illustrated in Fig. 2, the structures of GlCS genes are relatively complex, with the number of introns ranging from four to 16 across the family members. Notably, GlCS4 and GlCS6 lack untranslated regions (UTRs), whereas the other members possess both 5′ and 3′ UTRs. Among the GlCS gene family members, GlCS1, GlCS2, GlCS3, GlCS5, GlCS7, and GlCS8 were confirmed as full-length sequences, containing complete coding regions and intact 5′/3′ untranslated regions (UTRs). In contrast, GlCS4 and GlCS6 lack annotated UTRs, which could result from: (1) Technical truncation during sequencing/assembly, (2) genuine biological absence of UTRs in these transcripts, or (3) incomplete genome annotation. Further experimental validation (e.g., RACE assays) is required to determine their transcriptional boundaries.

Figure 2 The conserved motifs and gene structures of GlCS genes are based on phylogenetic relationships.

(A) Phylogenetic tree, (B) motifs compostitions, (C) gene structures.

A comparative analysis of the conserved protein motifs across the GlCS gene family revealed that the evolutionary relationships are largely consistent with their motif compositions and spatial distributions. GlCS2, GlCS5, and GlCS4, which cluster together on the phylogenetic tree, share Motif 1, Motif 2, Motif 3, Motif 4, Motif 5, and Motif 8. GlCS7 contains Motif 1, Motif 2, Motif 3, Motif 4, Motif 5, Motif 6, Motif 8, and Motif 10. GlCS1, GlCS3, GlCS6, and GlCS8, which form another clade, share Motif 1, Motif 3, Motif 4, Motif 6, Motif 7, Motif 9, and Motif 10. These findings highlight the conserved nature of certain motifs within specific clades, reflecting their functional and evolutionary significance.

A phylogenetic tree in Fig. 3 was generated with CS genes from G. lucidum and CS genes from other fungi to study their evolutionary relationships. This analysis revealed that fungal CS genes exhibit high similarity and are distributed across five distinct clades. GlCS3 clusters with Leucoagaricus sp. SymC.cos CS5 in the same clade. GlCS6 and GlCS8 are in the same clade as Leucoagaricus sp. SymC.cos CS1. GlCS1 is grouped with Leucoagaricus sp. SymC.cos CS1. GlCS7 forms a clade with Pleurotus ostreatus CS2. GlCS5 clusters with Leucoagaricus sp. SymC.cos CS3. GlCS2 is closely related to Leucoagaricus sp. SymC.cos CS2. GlCS4 is in a clade with Leucoagaricus sp. SymC.cos CS4.

Figure 3 Phylogenetic relationships of CS genes in Ganoderma lucidum and other fungal species.

Cis-acting elements in the GlCS proteins

Through the analysis of cis-acting elements in the promoter regions of eight chitin synthase genes (GlCS1-GlCS8) in G. lucidum (Fig. 4), we observed significant variations in both quantity and type of regulatory elements. A total of 47 functionally distinct regulatory elements were identified, primarily categorized into four major groups: hormone responsiveness, growth and development, stress responses, and light responses.

Figure 4 Cis-acting elements in the GlCS proteins of G. lucidum.

(A) Differently coloured histograms represent the sum of the four types of cis-acting elements in the GlCS gene; (B) heatmap of the cis-acting elements in each GlCS gene, with different colours indicating the quantity of cis-acting elements and white representing the absence of cis-acting elements.

Elements related to growth and development were universally present. All genes contained core promoter elements such as CAAT-box (14–24 copies) and TATA-box (3–16 copies), with GlCS7 showing the highest number of CAAT-box elements (24 copies) and GlCS3 exhibiting the most abundant TATA-box elements (16 copies), indicating active basal transcriptional activity of these genes.

ABA-responsive elements (ABRE and its variants) were highly enriched in GlCS2 (14 copies), GlCS6 (18 copies), and GlCS3 (nine copies). JA-responsive elements (TGACG/CGTCA-motif) showed prominent abundance in GlCS2 (16 copies), GlCS4 (16 copies), and GlCS3 (14 copies), suggesting precise regulation of these genes by phytohormones. All genes contained STRE heat stress-responsive elements (one to five copies), with GlCS2 showing the highest count (five copies). Light-responsive elements (G-box, GT1-motif, etc.) were significantly enriched in GlCS6 (12 G-box copies) and GlCS2 (eight G-box copies).

These findings reveal both conserved features and functional diversification among G. lucidum chitin synthase gene family members in terms of basal transcription, hormonal regulation, and environmental adaptation, providing important clues for understanding their functional differentiation.

Expression patterns of the GlCS genes during different developmental stages

From Fig. 5, it can be seen that the expression trends of the eight G. lucidum chitin synthase genes (GlCS) across different periods. The expression levels of GlCS1, GlCS2, GlCS5, GlCS6, and GlCS8 during the EA stage were significantly greater than those in other developmental stages, suggesting that these genes may play important roles in the expansion of the fruiting body. The expression level of GlCS3 during the MA stage was significantly greater than that observed during the other stages, indicating its potential specific function in the maturation of G. lucidum spores. Additionally, the expression level of GlCS7 during the PR stage was significantly greater than that in other stages, suggesting that this gene may be involved in regulating the formation of primordia.

Figure 5 Expression profiles of GlCS genes in various developmental stages.

Note: bars represent the mean values of three technical replicates ± SE. MY, mycelial; PR, primordium; EA, early fruiting body; MA, mature fruiting body. Different lowercase letters indicate significant differences (P < 0.05). Different capital letters indicate extremely significant differences (P < 0.01).

Expression patterns of the GlCS genes under heat stress

From Fig. 6, it can be observed that the expression trends of the eight G. lucidum chitin synthase genes (GlCS) detected by quantitative real-time PCR (RT-qPCR) under normal temperature and 40 °C conditions. Six genes, GlCS1, GlCS3, GlCS4, GlCS6, GlCS7, and GlCS8, were significantly upregulated under 40 °C high-temperature stress. This expression pattern indicates that these genes may collectively participate in the mechanism used by G. lucidum to respond to high-temperature stress. Their functions may involve maintaining cell wall integrity and activating related protective mechanisms, thereby increasing the survival ability of G. lucidum under adverse conditions.

Figure 6 Expression profiles of of eight GlCS genes in Ganoderma lucidum under heat stress.

Note: bars represent the mean values of three technical replicates ± SE. MY, mycelial at 25 °C; MY-40: themycelial incubated for 6 h at 40 °C. **** indicate extremely significant differences (P < 0.0001).

Discussion

The fungal cell wall functions as an external skeleton, providing both structural protection against environmental stresses and the mechanical force required for host invasion (Free, 2013). As a major structural component, chitin combines with glucans and mannans to form this essential cellular framework (Ruiz-Herrera & Ruiz-Medrano, 2004). Chitin synthase, which is the key enzyme in chitin biosynthesis, belongs to a multigene family in fungi. This study identified a total of eight CS gene family members from the genomic data of G. lucidum. Compared with S. cerevisiae (which contains only three CS genes), G. lucidum has a greater number of CS genes, which is comparable to the number of genes found in Aspergillus nidulans (which contains eight CS genes) but fewer than the number of genes in Flammulina velutipes (which contains nine CS genes). The GlCS proteins exhibited notable diversity in their physicochemical properties, including amino acid length, molecular mass, predicted isoelectric points, and stability profiles. All members of the GlCS protein family contain 5–7 transmembrane helical domains and exhibit plasma membrane localization characteristics, which is consistent with the traditional functional localization of chitin synthases. The endoplasmic reticulum-targeting signals observed in GlCS1, GlCS4, GlCS6 and GlCS8 may suggest their involvement in regulating the endoplasmic reticulum-Golgi secretory pathway. The nuclear localization signals predicted in GlCS3 and GlCS6 indicate these proteins may possess non-canonical functions such as gene expression regulation or nuclear membrane remodeling. Particularly noteworthy is the unique dual-targeting feature of GlCS6 to both peroxisomes and the mitochondrial inner membrane, which may be associated with its specialized role in oxidative stress response or energy metabolism. The predicted vacuolar-specific localization of GlCS8 suggests that vacuoles, as degradation and storage centers in fungal cells, may regulate chitin synthase activity by controlling the stability of GlCS8. This finding also implies that vacuoles may participate in the storage and recycling of chitin precursors or products. The observed variations imply that these gene members may fulfill specialized roles under different environmental conditions to support the varied physiological demands imposed throughout the developmental stages of G. lucidum.

A phylogenetic analysis revealed distinct functional groupings among the GlCS genes. GlCS4 clustered with S. cerevisiae CS1, which is known to function in cell separation repair. GlCS2 and GlCS5 formed a clade with S. cerevisiae CS2, the primary septum chitin synthase (Henar Valdivieso, Durán & Roncero, 1999). GlCS7 grouped with A. nidulans CHSB, a regulator of the hyphal growth rate, sporulation, and wall development (Lee et al., 2004). CHSB was constitutively expressed throughout fungal development and participates in both sexual (primordium formation, Hulle cell development, and ascospore maturation) and asexual growth processes. The three chitin synthase genes (chsA, chsB, and chsC) in A. nidulans exhibit distinct yet complementary roles in fungal development and stress adaptation. chsA is specialized for asexual reproduction, showing high expression in conidiophores and conidia while being negligible in sexual stages, suggesting its primary function in conidiation. chsC plays a key role in sexual development, particularly during ascospore maturation and cleistothecium formation, with stage-specific expression patterns, and also contributes to osmotic stress resistance alongside chsA. In contrast, chsB is ubiquitously expressed across all developmental stages and growth conditions, functioning as a scaffold for general cell wall organization rather than stress responses or specific morphological structures. While chsA and chsC show functional redundancy in asexual reproduction and stress adaptation, their specialized temporal and spatial expression patterns, combined with the constitutive role of chsB, highlight a coordinated division of labor in chitin biosynthesis that supports both fungal development and environmental adaptation. GlCS6 and GlCS8 were phylogenetically associated with A. nidulans CSmA, which is a key factor for maintaining cell wall integrity under low osmotic pressure (Yamada et al., 2005). GlCS1 and GlCS3 aligned with S. cerevisiae CS3, which synthesizes the majority of cellular chitin during division, particularly bud ring formation and cell wall distribution (Shaw et al., 1991; Gohlke, Muthukrishnan & Merzendorfer, 2017).

Cis-regulatory elements constitute a class of specialized DNA sequences that are positioned adjacent to structural genes and function as critical determinants of transcriptional control systems (Jia et al., 2023). These elements play a pivotal role in modulating the precision and efficiency of gene expression.Within genomic architecture, promoter regions frequently harbor diverse cis-regulatory motifs capable of interacting with sequence-specific DNA-binding proteins. Such molecular interactions establish sophisticated regulatory networks that govern spatiotemporal expression profiles (Kim & Wysocka, 2023). This sophisticated control system facilitates context-dependent gene activity across distinct biological contexts, including tissue-specific development, environmental adaptation, and stress response pathways, ultimately shaping cellular homeostasis and organismal fitness (Bell-Pedersen, Dunlap & Loros, 1996). The transcriptional regulation of yeast ribosomal protein gene TCM1 is coordinately promoted by atypical cis-acting and trans-acting elements (Hamil, Nam & Fried, 1988). In Neurospora crassa, the cis-acting element ACE is capable of driving the circadian regulation of eas (ccg-2) gene expression. All eight chitin synthase genes (GlCS1-GlCS8) contain cis-acting elements responsive to hormones, growth and development, stress responses, and light signaling. ABRE and its variants (ABRE3a/ABRE4) are highly enriched in GlCS2 (14 copies), GlCS6 (18 copies) and GlCS3 (nine copies), suggesting these genes may respond to osmotic stress or developmental regulation through the ABA pathway. In contrast, GlCS1 and GlCS8 nearly lack ABRE elements, implying their ABA-independent regulation. STRE (heat stress response element) is enriched across all eight chitin synthase genes, indicating these genes may collectively enhance G. lucidum’s adaptability to high temperature stress.

The fungal cell wall undergoes dynamic physical and chemical modifications—termed cell wall remodeling—to regulate its plasticity during growth, development, and morphogenesis (De Nobel, Van Den Ende & Klis, 2000). As a key structural component, chitin plays a central role in this process (Riquelme et al., 2018), enabling mechanical stress adaptation and controlled elongation processes (Zhou et al., 2019). Studies conducted across different fungal species have demonstrated the involvement of chitinases in diverse biological processes. In Pleurotus ostreatus, nine putative chs genes were identified, four of which are unique to basidiomycetes. The study found that knockout of chsb2-4 genes resulted in sparse mycelium, rough surface morphology, shorter aerial hyphae, and increased sensitivity to cell wall and membrane stress, accompanied by thinning of the cell wall. Although the relative content of chitin and β-glucan remained unchanged, other chitin and glucan synthase genes exhibited upregulation. These findings indicate that chsb2-4 genes play a distinct role in the structural formation of aerial hyphae and the cell wall (Schiphof et al., 2024). The deletion of Aspergillus nidulans ChiA causes a reduced hyphal growth rate (Takaya et al., 1998). A. nidulans AnChsB knockout leads to impaired conidiation, swollen hyphal tips, and abnormal cell walls (Jin, Iwama & Horiuchi, 2023). The deletion of Neurospora crassa NcChs-IRIP results in growth defects and increased drug sensitivity (Yarden & Yanofsky, 1991). Chitin synthases are crucial transmembrane enzymes that mediate chitin biosynthesis in fungi through their zymogenic activation mechanism. These enzymes perform multiple physiological functions including morphogenesis (cell wall assembly and hyphal development), autolysis (cell wall remodeling), nutrient assimilation, and fungal pathogenesis. In yeasts, they regulate critical processes like cell separation and sporulation, while in filamentous fungi they govern hyphal growth and virulence. Characterized as membrane-integrated glycosyltransferases, chitin synthases exhibit class-specific functional specialization and require proteolytic activation of their zymogen forms for proper activity. In this study, the expression patterns of eight GlCS genes in G. lucidum across different developmental stages were analyzed, which revealed significant differences among their expression levels. GlCS7 exhibited significantly greater expression during the PR stage than during the other stages, suggesting that this gene may play a key regulatory role in the transition from hyphae to fruiting body primordia. Gene silencing experiments involving Cordyceps militaris revealed that the Chi1 and Chi4 knockdown strains exhibited a complete inhibition of the primordium differentiation process, with the fruiting body yield decreasing by approximately half relative to that of wild-type controls (Zhang et al., 2023). Furthermore, the lengths of mature fruiting bodies in the Chi1- and Chi4-silenced strains were reduced by 27% and 38%, respectively. GlCS7 belongs to Class III chitin synthases, which in other fungi have been associated with specialized cell wall remodeling during developmental transitions. We hypothesize that GlCS7 may contribute to establishing the unique chitin architecture required for primordium morphogenesis, possibly through: (1) synthesis of specialized chitin microfibril arrangements, (2) interaction with specific glucan synthases to form composite wall structures, or (3) response to developmental signaling cascades. GlCS1, GlCS2, GlCS5, GlCS6, and GlCS8 presented the highest expression levels during the EA stage, indicating that these genes may play critical regulatory roles in the expansion of the fruiting body cap during its growth. GlCS3 was significantly more highly expressed during the MA stage than in the other developmental stages, suggesting its potential specific function in the maturation of G. lucidum spores. This finding aligns with research on chitin metabolism in filamentous fungi. For example, treatment with chitinase inhibitors suppresses hyphal division, preventing the normal production of arthrospores. The coordinated action of exochitinases and β-1,3-glucanases in N. crassa facilitates conidial dispersal through enzymatic degradation of the interconidial septum (Sándor et al., 1998). These results further support the potential role of GlCS3 in spore maturation and provide important clues for elucidating the molecular mechanisms underlying spore development in G. lucidum. In this study, we systematically characterized the expression profiles of the GlCS gene family through bioinformatics predictions and quantitative fluorescence analysis. To gain deeper insights into the biological functions of GlCS genes, we propose to conduct the following functional validation studies. The construction of GlCS gene knockout mutants using CRISPR-Cas9-mediated gene editing will directly reveal loss-of-function phenotypes. For example, knockout of GlCS7 may significantly delay or completely block primordium formation. Additionally, gene-specific silencing via RNA interference (RNAi), combined with a gradient concentration induction system, will enable precise investigation of gene dosage effects. By modulating GlCS3 expression levels and observing its impact on spore maturation, we can elucidate its quantitative regulatory role in spore development. These systematic functional studies will provide definitive experimental evidence, not only validating the predictions of this study but also uncovering the molecular mechanisms by which the GlCS gene family regulates morphogenesis, environmental adaptation, and developmental processes in G.lucidum. We have incorporated these experimental approaches into our follow-up research plan, which we believe will significantly advance the understanding of chitin synthesis regulatory networks in G.lucidum.

The optimal growth temperature range for G. lucidum is 25–30 °C. Elevated ambient temperatures exceeding 30 °C significantly inhibit fungal development, impairing both vegetative mycelial expansion and reproductive fruiting body differentiation (Zhang et al., 2021). Previous research has shown that in G. lucidum under low-nitrogen stress, the transcription factor GCN4 directly activates the expression of SWI6B. SWI6B then regulates the expression of chitin synthase, leading to cell wall thickening and polysaccharide accumulation. This indicates that the chitin synthesis pathway is closely related to the response of G. lucidum to environmental stress (Shi et al., 2025). This study investigated the expression responses of GlCS genes under heat stress to explore the role of the GlCS family in maintaining cell wall integrity and thermotolerance. When the environmental temperature was increased to 40 °C, the expression levels of six genes, GlCS1, GlCS3, GlCS4, GlCS6, GlCS7, and GlCS8, were significantly increased in G. lucidum, suggesting that these genes may participate in the heat stress defense mechanism. As chitin is an essential structural component of fungal cell walls, upregulated chitin biosynthesis likely compensates for thermal damage to the wall structure and maintains an appropriate degree of wall plasticity, thereby preventing cell lysis from being performed under high-temperature stress. Future studies could use gene knockout or overexpression techniques to validate the specific functions of GlCS genes in maintaining thermotolerance. This research provides important theoretical foundations for the cultivation of stress-resistant G. lucidum and the breeding of heat-tolerant strains.

Conclusions

In G. lucidum, eight chitin synthase (GlCS) family members have been identified and phylogenetically classified into five distinct classes. Our investigation revealed the stage-specific expression patterns of the GlCS genes throughout the fungal development process. In the EA stage, GlCS1, GlCS2, GlCS5, GlCS6, and GlCS8 exhibited significantly greater expression levels than they did in other developmental phases. In the MA stage, GlCS3 was predominantly expressed. In the PR stage, GlCS7 exhibited peak expression levels. Six genes (GlCS1, GlCS3, GlCS4, GlCS6, GlCS7, and GlCS8) were markedly upregulated under 40 °C thermal stress, suggesting their potential roles in thermotolerance mechanisms. These findings demonstrate functional diversification among the GlCS family members across different developmental stages and stress conditions. The observed differential expression patterns strongly indicate that each GlCS member has evolved specialized biological functions, particularly for mediating critical developmental transitions, facilitating environmental adaptations, and orchestrating dynamic cell wall remodeling processes. These findings significantly advance our understanding of how chitin synthase genes coordinate fungal growth and stress responses at the molecular level. Our study provides new insights that could be used in future studies to clarify the functions of GlCS genes.

Supplemental Information

Supplemental Information 1 MIQE checklist

Supplemental Information 2 Raw data of gene relative expression by qRT-PCR

Additional Information and Declarations

Competing Interests

Author Contributions

Data Availability

The authors declare that there are no competing interests.

Linling Liu conceived and designed the experiments, performed the experiments, analyzed the data, prepared figures and/or tables, authored or reviewed drafts of the article, and approved the final draft.

Yiming Yang conceived and designed the experiments, analyzed the data, authored or reviewed drafts of the article, and approved the final draft.

Jintao Li performed the experiments, analyzed the data, prepared figures and/or tables, and approved the final draft.

Yanliang Gao performed the experiments, analyzed the data, prepared figures and/or tables, and approved the final draft.

Meixia Yan conceived and designed the experiments, analyzed the data, authored or reviewed drafts of the article, and approved the final draft.

The following information was supplied regarding data availability:

The raw data is available in the Supplemental File.

The genomic data of G. lucidum are available at JGI Mycocosm: 260125-1 and GenBank: GCA_000271565.1.

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
