# Peer review of "Analysis of the chitin synthase gene family in Ganoderma lucidum: its structure, phylogeny, and expression patterns"

_PeerJ, doi:10.7717/peerj.20302_

## Round 0.1 · original submission · Major Revisions

Thank you for submitting your manuscript entitled "Analysis of the chitin synthase gene family in Ganoderma lucidum: Its structure, phylogeny, and expression patterns" to PeerJ. Your study provides a comprehensive genome-wide analysis of chitin synthase genes in Ganoderma lucidum, with insights into their expression during developmental stages and under heat stress. The manuscript is clearly written and contains high-quality figures and data.

After evaluation by two expert reviewers, the consensus is that while the manuscript presents valuable findings, it requires major revisions to improve its scientific rigor and completeness. Below is a summary of the key points that must be addressed.

Major Points and Required Revisions:

Lack of Functional Validation

As noted by Reviewer 2, the manuscript relies entirely on in silico and transcript expression data without functional assays. This limits the ability to establish a causal relationship between GlCS gene expression and specific developmental or stress-related roles.

Required revision: Clearly acknowledge this limitation in the Discussion. Discuss how future functional validation (e.g., gene knockout, RNAi, localization studies) could clarify gene roles in morphogenesis or stress response.

Neglect of Chitinase Gene Interactions

Reviewer 1 highlighted the omission of chitinases, which often act in concert with chitin synthases during fungal cell wall remodeling.

Required revision: Include a paragraph in the Discussion addressing the possible interaction between chitin synthases and chitinases, referencing findings from other fungal species. Indicate whether any preliminary data or plans exist for investigating chitinase gene expression in G. lucidum.

Incomplete Methodological Details

The manuscript lacks accession numbers and metadata for sequences used in phylogenetic analyses. The rationale for selecting three S. cerevisiae chitin synthase genes as queries is also unclear.

Required revision:

Provide accession numbers and full details for all fungal CS sequences used in the phylogenetic tree.

Explain the criteria used to select the S. cerevisiae CS genes as BLAST queries.

Include raw RT-qPCR data (e.g., Ct values) and indicate whether primer efficiencies were tested.

Biological Interpretation of Subcellular Localization

Several GlCS proteins are predicted to localize to unusual organelles (e.g., vacuoles, mitochondria, peroxisomes). These were not discussed in the manuscript.

Required revision: Add a short discussion interpreting these findings—whether they may suggest divergent functions or reflect prediction limitations.

Minor Corrections and Formatting

Reviewer 2 noted typographic inconsistencies such as unitalicized species names.

Required revision:

Ensure all species names (e.g., Ganoderma lucidum, Saccharomyces cerevisiae) are italicized throughout.

Clarify figure legends and table titles, particularly for Table 3.

Consider minor grammatical edits to enhance clarity, though these do not affect scientific content.

Summary
This is a promising manuscript that presents novel insights into the G. lucidum chitin synthase gene family. With the revisions outlined above, your work will be significantly strengthened and more suitable for publication. Please submit:

A point-by-point response to all reviewer comments, indicating how each was addressed.

A revised manuscript with tracked changes and a clean version.

We look forward to receiving your revised submission.

Reviewer 1 ·

Basic reporting

Using bioinformatics, authors detected up to eight chitin synthase (CS) genes in Ganoderma lucidium, an ancient medicinal basidiomycete fungus, by retrieving three CS genes from Saccharomyces cerevisiae as the NCBI database. Following expression of GlCS they analyzed the structure and potential function of GlCS transcripts and also some properties of proteins encoded by these transcripts such as transmembrane helices and plasma membrane localization. Based on the level of GlCS transcription, they observed a stage-specific function of GlCS basically during fruiting body development. All 8 genes contained cis-acting elements related to responses to hormones, growth and development, stress development and light responses. Six out of these genes were upregulated at 40 °C suggesting they are responsible for thermotolerance of G. lucidium.
Authors revised sufficient and pertinent references and their work is well sustented. They pofessionally structured their work which contained acceptable quality tables and figures. Though they did not present a formal hypothesis they assumed, based on previous publications, that several CS genes should participate during development of this medicinal basidiomycete.

Experimental design

Authors used morphogenesis of fruiting bodies and cellular thermostress of G. lucidium as models to conduct the study the role of CS genes in the control of these processes. To this aim, they used standard molecular biology protocols adapted to specific objectives of their work. Personally, I did not register innovations/improvements to these methods. Therefore, I do not have much to comment in this section.

Validity of the findings

This submission contains valid and robust information. Apart from the well-studied Aspergillus nidulans, S. cerevisiae and other less known organisms, information on the presence of several forms or isoforms of CS and their specific functions is not abundant. This fact confers impact and novelty of this study particularly because it was carried out in a poorly known fungus except in some Asia countries where it has been used in medicine for over 2000 years. Besides, authors theorize (see comments below) the specific stage of development the genes are operational and open the possibilty to improve cultivation of a medically important organism. The latter therefore entails some benefit described in the last paragraph of the submission.
Conclusions are fine and well described and do not exceed their results. No speculations are made (see comments below).

Additional comments

1. What about the role of chitinases?. You mention these enzymes in lines 352-353 but not further precisions are made. In general, it is well documented that these hydrolases function, at least in fungi and in vivo, in parallel with Cs´s to give "shape" the polymer being synthesized and also participate in the formation of chitin microfibrils operating, in a simplistic language, as "seamstress who cuts and pastes" or "who does alterations and repairs". I fully understand that yours is a bioinformatic analysis. But also believe that results presented are in way only half of the story (not invalid) if chitinase genes are not explored along. What would it happen if a particular CS gene is upregulated in a specific stage?. A mate chitinase gene should be downregulated or also upregulated?. Do you have a comment?
2. What does it mean that some GlCS genes have nuclear, peroxisome or mitochondrial inner membrane transmembrane signals and why is it interesting that GlCS8 localizes to vacuoles?.
3. What can you comment about zymogenicity of most chitin synthases?.

Reviewer 2 ·

Basic reporting

This manuscript is well organized with a sufficiently information in the introduction section and high-quality figures that support the study's findings. However, a significant limitation is the absence of raw sequence data and accession number of sequences used in silico analysis, for example in phylogenetic analysis, which has not been provided. For the grammatically issue, please correct the species name to be italicized through out the manuscript.

Experimental design

The hypothesis of this study is well defined to evaluate the function of 8 chitin synthase gene in G. lucidum. However, the major issue with the manuscript is its insufficient experimental design. The results contain only in silico analysis and chitin synthase expression profiles but lack of experimental data, for example cell wall integrity evaluation in each stage, to support and describe relationship between expression profile and cell morphology. In addition, adding the hypothesis and reason why select 3 chitin synthase genes from S. cerevisiae as the query for BLAST search against G. lucidum genome in gene mining process.

Validity of the findings

Due to lack of experimental data in the filed related with cell wall integrity and/or chitin content, therefore this could be difficult to propose and conclude the relationship of expression level of 8 chitin synthase genes in each cell development stages. Importantly, the manuscript would benefit if the authors add more comprehensive discussion of the findings.

---

## Round 0.2 · Minor Revisions

Dear Dr. Liu,

Please carefully address the reviewer comments, particularly regarding species names and other typos and small mistakes along the manuscript.

Reviewer 2 ·

Basic reporting

The revised manuscript is well organized and has been edited regarding reviewer's suggestions. The information in literature review is sufficient. However, the grammatically errors, for example species names, need to be rechecked and edited though out the manuscript.

Experimental design

The hypothesis and reason why select 3 chitin synthase genes from S. cerevisiae as the query for BLAST search against G. lucidum genome has been described and added in this revised manuscript.

Validity of the findings

Due to insufficient of experimental data related with cell wall integrity and/or chitin content evaluation in each stage to describe expression profiles of 8 chitin genes in different cell stage. However, the authors already added more comprehensive discussion in the manuscript.

Additional comments

The point-by-point response to reviewer's and editor's comments has been added in this revised version. However, grammatically errors, for example species name, need to be addressed before publish.

---

## Round 0.3 · accepted · Accept

Congratulations on the acceptance of your manuscript.